# Professional Quality of Life of Healthcare Workers in Hospital Emergency Departments

**DOI:** 10.3390/bs12060188

**Published:** 2022-06-13

**Authors:** Daniel Pérez-Valdecantos, Alberto Caballero-García, Hugo J. Bello, David Noriega-González, Nora Palomar-Ciria, Alba Roche, Enrique Roche, Alfredo Córdova-Martínez

**Affiliations:** 1Department of Biochemistry, Molecular Biology and Physiology, Faculty of Health Sciences, GIR “Physical Exercise and Aging”, University of Valladolid, Campus “Los Pajaritos”, 42004 Soria, Spain; danielperezvaldecantos@gmail.com (D.P.-V.); alba.1078@gmail.com (A.R.); 2Department of Anatomy and Radiology, Faculty of Health Sciences, GIR “Physical Exercise and Aging”, University of Valladolid, Campus “Los Pajaritos”, 42004 Soria, Spain; alberto.caballero@uva.es; 3Department of Mathematics, School of Forestry, Agronomy and Bioenergy Engineering, GIR “Physical Exercise and Aging”, University of Valladolid, Campus “Los Pajaritos”, 42004 Soria, Spain; hjbello.wk@gmail.com; 4Department of Surgery, Ophthalmology, Otolaryngology and Physiotherapy, Faculty of Medicine, Hospital Clínico Universitario de Valladolid, 47003 Valladolid, Spain; davidcesar.noriega@uva.es; 5Psychiatry Service, Complejo Asistencial de Soria, 42005 Soria, Spain; npalomar@saludcastillayleon.es; 6Instituto de Bioingeniería y Departamento de Biología Aplicada-Nutrición, Universidad Miguel-Hernández, 03202 Elche, Spain; eroche@umh.es; 7Instituto de Investigación Sanitaria y Biomédica de Alicante (ISABIAL), 03010 Alicante, Spain; 8CIBER Fisiopatología de la Obesidad y la Nutrición (CIBEROBN), Instituto de Salud Carlos III (ISCIII), 28029 Madrid, Spain

**Keywords:** emergency, health professionals, quality of life, self-efficacy, sleep quality, stress

## Abstract

In previous publications, we have reported that professionals in emergency departments undergo high levels of stress according to the amounts of salivary biomarkers (α-amylase and cortisol). The stress seems to be counteracted by increased levels of dehydroepiandrosterone. This hypothesis is confirmed in the answers to different tests indicating no working stress, no anxiety, optimal self-efficacy, and good sleeping quality. Altogether, these previous results suggest an optimal adaptation of these workers to the demanding situations that occur in emergency departments. To complete this research, we decided to evaluate the quality of life of health professionals working in the emergency departments of two Spanish hospitals. A descriptive cross-sectional study was carried out during the pre-pandemic months of July and August 2019. We determined the professional quality of life through the QPL-35 questionnaire in 97 participants, including nurses (n = 59) and medical doctors (n = 38). Answers to the test indicate that the studied participants working in emergency departments have a good perception of professional quality of life. This is reflected in the three dimensions of the questionnaire: managerial support, workloads, and intrinsic motivation. Based on the results of all answered tests, emergency healthcare staff seem to have adequate professional management, with levels of stress, sleep, and quality of life in line with a controlled lifestyle. Altogether, this would allow a correct adaptation to the demanding situations experienced in emergency departments. The relevance to clinical practice is that the COVID-19 pandemic has disrupted this controlled professional management.

## 1. Introduction

The Spanish health system is an open institution that establishes a continuous relationship between the organization’s structure, objectives, people, environment, and resources, adapting to the social needs that demand well-being and quality of life. According to the World Health Organization (WHO), quality of life is defined as “the individual’s perception of his or her position in life within the cultural context and value system in which he or she lives and with respect to his or her goals, expectations, norms and concerns”. Healthcare workers are confronted on a daily basis with complex tasks that are influenced by various stressors involving emotional problems. This is related to the organization of work and could affect negatively their physical and mental health [1].

In psychology, there are different instruments to evaluate the state of human resources and to guide intervention strategies to improve the quality of healthcare workers. In this sense, the model of Human System Audit (HSA), or Human Analysis System [2], proposes a systemic model that includes organizational behaviour, with instruments for evaluating the dimensions of the organization and human behaviour, and a management control system. This model includes, as part of the organization, the psychosocial processes that the worker may undergo and that may influence performance and work results [2,3].

In this sense, work stress affects negatively the quality of life and health of affected people. This results in a state of moderate anxiety causing difficulty in falling asleep [4]. The concept of quality of life is complex and the construct incorporates physical, mental, and emotional elements, in an attempt to reflect the holistic sense of health [5]. The deterioration of quality of life may be the result of heterogeneous conditions and the interaction between them [6]. For instance, disorders in sleep quality are associated with disorders in quality of life, which are accentuated in professions with occupational risk factors and with night shift work schedules, a situation that could diminish work performance [7,8]. In this context, sleep is necessary for the maintenance of a large number of psychological and organic functions. Emotional reactions such as anxiety may be one of the most disturbing, both for getting to sleep and for maintaining a restful sleep [9,10]. The insomnia consequences on quality of life can be significant: daytime sleepiness, decreased work performance, mood changes, impaired interpersonal relationships, and increased risk of accidents [9,10]. However, if poor sleep does not cause problems for work performance, pharmacological treatment is not necessary [11]. A sleep problem is a risk factor for psychological disorders such as depression, anxiety, and suicide [12]. Sleep in adequate quantity and quality provides the recovery of physical and mental well-being and improves mood, concentration, and memory [13].

On the other hand, both work performance and self-efficacy are important, to improve efficiency, motivation, and work attitude [14,15]. Self-efficacy refers to the feeling of one’s own ability, sensitivity, and prudence [16]. People with a strong sense of self-efficacy have the courage to overcome difficulties and show good emotional and behavioural states. In this sense, anxiety is a mediator between academic performance and self-efficacy [17]. Subjects with low self-efficacy generate greater anxiety, resulting in repercussions on low academic performance. People with high self-efficacy and perceived control have lower cardiovascular reactivity [18]. On the contrary, subjects who underestimate their capabilities increase their physiological response by giving up when performance becomes more complicated [19].

There are numerous publications about stress and burnout, regarding alterations in psychological well-being or job satisfaction, sleep disorders, and the psychosocial consequences suffered by healthcare professionals in their work environment [20,21,22,23,24,25,26,27,28,29,30]. The situation during pandemics has highlighted the importance of the state of health and the quality of life of health professionals. The WHO established that health is a state of complete well-being that goes beyond the mere absence of disease/disorder [4]. Otherwise said, health should be considered more as a right than a necessity. The stress situation in emergency services has been exacerbated during the COVID-19 pandemic. Nevertheless, the present report refers to a pre-pandemic period, reflecting correct management of stress. Therefore, the aim of this work is to complete the aspects related to stress studied in previous reports [31,32] such as work stress, anxiety, self-efficacy, and sleeping quality, with the quality of life perception of healthcare workers (nurses and medical doctors) in emergency departments (EDs). These results indicate that the working pre-pandemic environmental structure of the Spanish health system seemed to be optimal for ED professionals. The results of the present report are part of the largest project (study of the psychological status of healthcare professionals working in emergency departments before pandemics) and complement previously published results [31,32].

## 2. Materials and Methods

### 2.1. Study Design

A descriptive cross-sectional study was conducted during the months of July and August 2019 in the EDs of two public hospitals representative of the Spanish Health System: Hospital Clínico Universitario de Valladolid (HCUV) and Hospital Santa Bárbara de Soria (HSBS). The subjects were all healthcare workers that developed their activity in either the morning or the afternoon shift. The project was approved by the Ethics Committee for Clinical Research of the Burgos Health Area (Reference: CEIC 1984).

### 2.2. Sample

Participation in the study was voluntary and received no economic compensation. Written informed consent was obtained from every subject, in accordance with protocols from the Ethics Committee CEIC (from Spanish: “Comité Ético de Investigación Clínica”). Anonymity was preserved and the subjects were informed that they could withdraw freely at any time. Inclusion criteria were fulfilled by all ED professionals in both hospitals: (i) healthcare professional (nurse or medical doctor) in the ED of HCUV and HSBS, (ii) good health condition, with no mental or physical pathology that would disqualify them from their profession, (iii) absence of endocrine or any other pathology, and (iv) 18 years or older.

A total of 97 participants were included: 59 nursing professionals (10 men and 49 women) and 38 medicine doctors (10 men and 28 women). Regarding the working schedule, the distribution was, 66 worked in the morning shift and 31 in the afternoon shift. The role of participants in the organization was, 27 were permanent personnel, 34 temporary substitutes, 20 interim, and 14 professionals in formation (MIR) (Table 1). The mean age of participants in the study is shown in Table 2.

### 2.3. Data Collection

Professional quality of life can be determined through the QPL-35 questionnaire proposed by the authors of [33] and based on the theoretical framework of [34]. The Spanish validated version is the CVP-35 questionnaire [35]. In this context, QPL-35 has already been validated in the Spanish population [36]. The QPL-35 is a multidimensional measure of professional quality of life. The questionnaire includes three dimensions: managerial support, workloads, and intrinsic motivation. Managerial support is related to the perception of the support received from the boss or leader, whose presence can be a motivating and safety factor at work or on the contrary, a factor of stress and discomfort that may generate tensions in the worker’s environment. Workloads refer to the perception that the worker has of the demands of the job being evaluated through 12 items. Workloads are related to the activities carried out at work, and can be quantitative (excess of activities to be carried out in a certain period of time or an excessive number of hours at work) or qualitative (excessive demand in relation to skills, knowledge level of the worker or level of responsibility in clinical decision making). Quantitative and qualitative workloads are associated with work stress and dissatisfaction. Intrinsic motivation includes a set of internal and external factors that partly determine a person’s actions. It can be internal (spontaneous need) or external (induced need). Finally, the perceived overall quality of professional life (unique item) is the whole personal perception about the quality of professional life of the individual. The questionnaire consists of 35 questions answered on a scale of 1 to 10 (1 meaning “not at all” and 10 meaning “very much”). The test was provided at the beginning of the working shift and collected at the end. The average time to answer the questionnaire was around 20–25 min.

### 2.4. Statistics

R software package, R-Studio, and Python (Pandas, Numpy) were used to analyse data. T-tests were used to measure statistical differences between means (significance level α = 0.05). Cohen’s d statistic was used to measure the effect size between groups. Quantitative variables were expressed as mean + standard deviation.

## 3. Results

We have recently published that emergency professionals display increased salivary levels of α-amylase and dehydroepiandrosterone (DHEA) during the working day. The pattern of these markers may suggest a counteracting mechanism of DHEA against the stress reflected by amylase increases [31]. In order to verify this hypothesis (low stress due to counteracting the action of DHEA), we have analysed different psychological aspects in the same group of healthcare professionals through different tests related to behaviours resulting from stress [31,32]. These include the Medical Personnel Stress Survey (MPSS-R) [37], the State-Trait Anxiety Inventory (STAI) [38], the self-efficacy test [39], and the sleeping quality questionnaire or COS (from Spanish “Cuestionario de Oviedo del Sueño”) [40]. We wanted to verify if all positive attitudes reflected in these tests could be summarized in optimal quality of professional life. Previously published results corresponding to MPSS-R, STAI, Self-Efficacy, and COS questionnaires are summarized in Table 3. MPSS-R indicates that measured stress levels were high for all staff groups studied. Somatic distress and organizational stress were the most prominent stress markers, followed by job dissatisfaction and negative attitudes towards patients [31]. The other tests displayed optimal results [32].

Following our published findings, the STAI questionnaire revealed no significant differences in the scales of State and Trait regarding gender, professional status, and hospitals. These results indicate that professionals in EDs do not display anxiety and they do not seem to be affected by the situations they have to undergo in ED [32]. In the same vein, the self-efficacy test displayed no significant differences, regardless of professional status, gender and hospital. This result suggests that the tasks at work are developed in an efficient and adequate way [32]. Finally, the COS questionnaire answers indicate acceptable sleep quality with no significant differences between the different groups of gender, professional status, or hospitals studied [32].

Table 4 shows the results corresponding to the items evaluated in relation to the quality of professional life (QPL-35 questionnaire). The different aspects analysed are shown in each column. We can observe that, in any of the dimensions, there are scarce differences regarding gender or hospital. In this context, the compared groups have low Cohen’s d values which implies small effect sizes, with the exception of men vs. women that present a medium effect size (Table 5).

Figure 1 shows the dispersion of the values corresponding to overall professional quality of life from the QPL-35 questionnaire. Additionally, in this case, there are no great differences between groups and we can find that dispersion is greater in medical staff and in HSB. The overall assessment of the professional quality of life was 5.6 (from a total of 10).

## 4. Discussion

In the health care field, there is a growing interest in relation to psychosocial aspects related to work environment and quality of life. This is a key question for ED professionals due to the limited situations they have to undergo suddenly. This population is at high risk of burnout, role conflict, and job dissatisfaction. This can lead to a deterioration of their quality of life. In this line, our group has already published a study showing high levels of stress in health professionals working in EDs [31]. The study was carried out during the pre-pandemic period, reflecting the usual work situation undergone in the EDs of the Spanish health system.

The results of our work show that stress and job dissatisfaction scores were high but accompanied by an adequate level of perceived sleep quality. In addition, the perception of self-efficacy and no anxiety was good in general [32]. Likewise, the workers studied had a high perception of responsibility, with no differences between gender, profession, or hospital. We did not find different appreciations of work demands according to professional category. It can be inferred from the results that perception of self-efficacy and responsibility are protecting factors for sleep disturbances, despite other work characteristics. This highlights the importance of the role of the psychosocial environment at work and the interrelationships between stress, the organizational system on the work development, and the health status of healthcare professionals.

Regarding the high level of stress presented by health professionals, we have previously reported the quantitative response to stress in these health professionals [31]. We have observed that stress, measured through cortisol and α-amylase, was evident in doctors and nurses, with an increase in DHEA, which, due to its anabolic condition, could counteract the effect of stress.

Previous research indicates that organizational and personal factors were associated with the work engagement of hospital staff [20]. The most important organizational predictors of work engagement were workload, values, and community. In addition, the quality of patient care and overall organizational well-being could be affected by the level of work engagement. In this context, the quality of work life among healthcare workers was measured by observing different results depending on the organizational environment [21]. In our work, although the workload was high, the effects would not have an impact on job performance, taking into account the high intrinsic motivation of the subjects studied in all professional categories.

When analysing the MPSS-R test results, which measures the level of stress directly influenced by organizational actions, it was observed that there was a high level of stress in healthcare personnel [31]. This level of stress seemed to be more dependent on negative organizational actions or on patient care dimensions. These results are interesting taking into account that only physically and psychologically healthy subjects were included in the study. We can discuss at this point if the lack of objective stress response is due to the fact that stress (measured with cortisol and α-amylase) could be compensated with an elevation of anabolic hormones such as DHEA [31]. Thus, this situation may indeed have a direct influence on work efficiency and on the quality of life perceived by the health professionals.

Previous reports observed that nurses tend to be the least satisfied with their work [22,23,24]. On the other hand, after factor analysis of the quality of life of health professionals working in the ED, the organizational structure obtained in our sample also coincides almost entirely with other studies [36], supporting the factorial validity of the instrument and the appropriateness of the three-dimensional structure proposed.

Regarding sleep, one of the prominent factors that has been found to be influential in health workers is shift work, which is also considered to be a public health problem that affects working and personal life [25]. Surprisingly, in our study, good subjective sleep quality is reported. This interpretation has to be done with caution because only daytime workers were included [32].

Self-efficacy, i.e., self-confidence in oneself of being able to perform the behaviour one proposes, allows overcoming difficulties and exhibiting good emotional and behavioural states [15,16,17,18]. However, high self-efficacy is not a sufficient condition for adequate performance. In previous research on the topic, it was posed that perceived inefficacy in coping with aversive situations may trigger anxiety [42]. We have observed high levels of self-efficacy in our sample [32]. Hence, these health professionals may perceive a better state of health or be able to have healthier lifestyle habits that make it easier to cope efficiently with their working life. Moreover, high perceived self-efficacy may protect from anxiety manifestations, according to some authors [42].

These overall results obtained in all tests (MPSS-R, STAI, Self-Efficacy, and COS questionnaires) [31,32] suggest that the ED professionals seem to have adequate work management and control of stress in the different situations they must face. We have suggested that a stable lifestyle and family habits could contribute to stress management. Adequate relationships between peers, participation in professional forums, and even maintaining friendships in the workplace could contribute to stress control and prevent the development of stress-related pathologies. This hypothesis has been verified with the QPL-35 questionnaire.

The study has some limitations such as the small sample size and the non-randomized sample. The likely explanation is that we have selected a number of subjects who could give a stronger response to our findings. It is true that some research carried out in similar populations has found significant relationships between sleep disorders, anxiety, and quality of life and that this is probably because they have taken into account other socio-health components that we have not included in this study.

## 5. Conclusions

Based on the results, the pre-pandemic situation indicates that ED healthcare staff have adequate professional management, with levels of stress, sleep, and quality of life in line with what could be indicated as a stable and controlled lifestyle. Therefore, the goal will be focused on establishing programmes to improve the quality of life of emergency healthcare professionals, targeted at skills, such as self-efficacy, sleep hygiene, and coping strategies, among others. This could result in better patient care.

## Figures and Tables

**Figure 1 behavsci-12-00188-f001:**
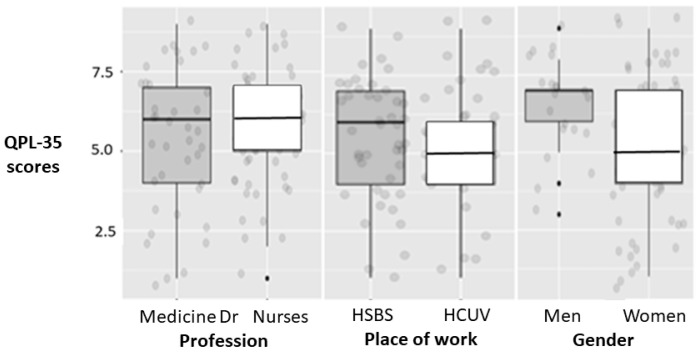
Dispersion of Professional quality of life (QPL-35 questionnaire) in the studied participants. Abbreviations used: HCUV (Hospital Clínico Universitario de Valladolid), HSBS (Hospital Santa Bárbara de Soria). Boxplots for the different groups showing medians and interquartile ranges.

**Table 1 behavsci-12-00188-t001:** Characteristics of the group of healthcare professionals studied.

Professionals	n	% (Gender)
n	97	20.6/79.4 (M/W)
Profession		
Nurses	59	60.1
Medical doctors	38	39.9
Place of work		
HCUV	45	46.4
HSBS	52	53.6

Abbreviations used: HCUV: Hospital Clínico Universitario de Valladolid; HSBS: Hospital Santa Bárbara de Soria; M/W: men/women.

**Table 2 behavsci-12-00188-t002:** Ages of participants.

Ages (Years)	
All participants	38.6 + 11.9
Nurses	39.0 ± 13.2
Medical doctors	39.6 ± 13.5
HCUV	34.7 ± 9.7
HSBS	42.4 ± 12.5
Men	39.9 ± 15.2
Women	39.5 ± 12.1

Abbreviations used: HCUV: Hospital Clínico Universitario de Valladolid; HSBS: Hospital Santa Bárbara de Soria.

**Table 3 behavsci-12-00188-t003:** Summary of scores (expressed as means) obtained from the different tests passed to ED professionals in previous studies by our group and the corresponding interpretation. For additional details regarding different dimensions of these scores, see [31,32].

Test	Score	Result
MPSS-R	65.6	High working stress
STAI state/trait	25.2/24.7	Low anxiety
Self-efficacy	29.3	High efficacy level
COS	4.1	Optimal subjective satisfaction of sleep

Abbreviations used: COS: sleeping quality questionnaire (from Spanish “Cuestionario de Oviedo del Sueño”); MPSS-R: Medical Personnel Stress Survey; STAI: State-Trait Anxiety Inventory.

**Table 4 behavsci-12-00188-t004:** Values obtained in the QPL-35 questionnaire.

	Dimension 1: Managerial Support	Dimension 2: Workloads	Dimension 3: Intrinsic Motivation	Overall Perceived Quality of Professional Life
Women	7.6 ± 1.9	6.5 ± 1.4	8.1 ± 0.9	5.4 ± 2.0
Men	7.2 ± 2.4	6.0 ± 2.1	7.5 ± 1.8	6.0 ± 2.3
Nurse	7.0 ± 2.0	6.3 ± 1.7	8.0 ± 1.3	5.8 ± 2.0
Medical Dr	7.6 ± 2.4	6.4 ± 1.8	7.8 ± 1.6	5.1 ± 2.3
HCUV	8.0 ± 2.1	6.4 ± 1.7	8.3 ± 0.9	5.5 ± 1.2
HSBS	7.2 ± 1.8	6.7 ±1.4	7.8 ± 1.0	5.5 ± 2.0
TOTAL	7.5 ± 1.9	6.8 ± 1.5	8.0 ± 1.0	5.6 ± 2.0

95% confidence intervals for the median calculated using bootstrap. Abbreviations used: HCUV: Hospital Clínico Universitario de Valladolid; HSBS: Hospital Santa Bárbara de Soria.

**Table 5 behavsci-12-00188-t005:** Effect size between groups according Cohen’s d analysis.

Group Comparison	Dimension 1: Managerial Support	Dimension 2: Workloads	Dimension 3: Intrinsic Motivation	Overall Perceived Quality of Professional Life
Men vs. Women	−0.06	0.27	−0.45	0.64
Medical Dr vs. Nurses	−0.06	0.44	−0.35	−0.25
HCUV vs. HSBS	0.03	−0.18	−0.08	0.08

Cohen’s d values lower than 0.2 imply a small effect size and d values between 0.5 and 0.8 imply a medium effect size [41]. Abbreviations used: HCUV: Hospital Clínico Universitario de Valladolid; HSBS: Hospital Santa Bárbara de Soria.

## Data Availability

Data are available from the corresponding author upon reasonable request.

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
