# Peer review of "Professional Quality of Life of Healthcare Workers in Hospital Emergency Departments"

_behavsci, 2022, doi:10.3390/bs12060188_

Round 1
Reviewer 1 Report
The article entitled “Health and Quality of Life of Healthcare Professionals Working in Hospital Emergency Departments” explores the important topic of stress and quality of life in healthcare workers. Overall, the manuscript is decently written and has the potential to contribute to the literature. One of the most significant strengths is the study’s applied sample including a variety of healthcare workers in multiple settings.
That being said, I would like to the see the authors address the following concerns:
1. I found the introduction difficult to follow. The authors raise the issue of the pandemic to highlight the study’s relevance, yet the data are from before the pandemic. The authors raise their own prior research on the efficacy of a medical treatment for stress, but this study actually has nothing to do with that treatment, and in turn, is not directly related. The introduction would benefit from citing the rich literature on workplace burnout and stress among healthcare workers.
2. The authors claim to study “professional management”, which far exceeds their data and method. In reality, the authors provide a single snapshot of self-reported quality of life by healthcare workers. The authors have no control group or baselines to compare their findings to. Their conclusion that healthcare workers do not appear to suffer from high levels of stress seems to be a stretch.
3. The authors need to elaborate on their sample inclusion criteria. How many people were excluded as a result of these criteria? Would eliminating anyone with absence of “any other pathology” possibly bias their results? The sample size is also quite small.
4. The tables need work. Tables should stand on their own with clear table notes. The tables include acronyms without definitions, which forces the reader to comb the manuscript to find the definitions. The tables could also benefit from improved formatting, such as horizontal lines that separate unrelated variables. For example, HCUV and HSBS refer to institutions and their %s add up to 100. The same goes for nurses and doctors. Yet the table presents them all as if they’re related.
5. The data collection is confusing. The authors discuss DHEA as part of their own recent publications, but this has nothing to do with the study as it is presented. This is not part of the inclusion criteria, the method, or the analyses.
6. It’s unclear why Python was required for such basic analyses. What was Python used for? R is more than capable of running t-tests and descriptive analyses.
7. What standard are the authors using to define low, medium, and large effect sizes? There are guidelines in the literature.
8. Some of the presented findings do not seem to correspond to any hypothesis. Why are there comparisons between professions, hospitals, and gender? The introduction does not lead the reader to expect these analyses.
Author Response
REVIEWER-1
The article entitled “Health and Quality of Life of Healthcare Professionals Working in Hospital Emergency Departments” explores the important topic of stress and quality of life in healthcare workers. Overall, the manuscript is decently written and has the potential to contribute to the literature. One of the most significant strengths is the study’s applied sample including a variety of healthcare workers in multiple settings.
That being said, I would like to the see the authors address the following concerns:
- I found the introduction difficult to follow. The authors raise the issue of the pandemic to highlight the study’s relevance, yet the data are from before the pandemic. The authors raise their own prior research on the efficacy of a medical treatment for stress, but this study actually has nothing to do with that treatment, and in turn, is not directly related. The introduction would benefit from citing the rich literature on workplace burnout and stress among healthcare workers.
ANSWER: We appreciate this important concern from the reviewer. It is true that the data refer to pre-pandemic situation, but following indications from the editorial team, we have mentioned this situation, because pandemics has affected particularly workers of Emergency Departments. We just mention that pandemics has disrupted this situation and the goal, at least in the Spanish Health System, is to return to the working conditions of the pre-pandemic state. On the other hand, we propose no medical treatment for stress. We only mention different situations that could contribute to stress, such as poor sleep quality, anxiety resulting from low self-efficacy and motivation. These constructs were collected in previous research and we think that the present report could culminate with the overall perception of quality of professional life. Finally, the search in PubMed introducing the following keywords. “burnout and stress and healthcare workers” revealed 534 reviews. Obviously, it is impossible to cited all this literature. However, we have cited some selected references according to Reviewer suggestions. See new References 26-30.
- The authors claim to study “professional management”, which far exceeds their data and method. In reality, the authors provide a single snapshot of self-reported quality of life by healthcare workers. The authors have no control group or baselines to compare their findings to. Their conclusion that healthcare workers do not appear to suffer from high levels of stress seems to be a stretch.
ANSWER: Quality of professional life is studied through the validated questionnaire QPL-35 (CVP-35 for the Spanish version). The questionnaire gives a score that according to the scientific literature gives an adequate quality of professional life for Emergency workers. Since QPL-35 is validated and widely used, the existence of a control group is not determinant in the present report. Nevertheless, it is complicate to define a basal situation or a control group in the environment of Emergency Departments. We know that some research groups have collected some data during pandemics. We plan to propose a future collaboration to compare our pre-pandemic data with the data obtained during pandemics. In any case, our data in the present report need to be published first in order to address this collaboration.
- The authors need to elaborate on their sample inclusion criteria. How many people were excluded as a result of these criteria? Would eliminating anyone with absence of “any other pathology” possibly bias their results? The sample size is also quite small.
ANSWER: We agree with the Reviewer that presence of pathologies in workers of Emergency Departments could be considered as a bias and this aspect can have a direct influence on results. However, all workers in the ED fulfil the inclusion criteria. In addition, we agree as well that our sample is very small, This is the eternal problem of funding in the Scientific Spanish System. We hope that the publication of these results will open new collaborations with other universities in order to afford a larger research in a near future. In any case, the number of participants that do not fulfil the inclusion criteria is indicated according Reviewer suggestion (see lanes 119-120).
- The tables need work. Tables should stand on their own with clear table notes. The tables include acronyms without definitions, which forces the reader to comb the manuscript to find the definitions. The tables could also benefit from improved formatting, such as horizontal lines that separate unrelated variables. For example, HCUV and HSBS refer to institutions and their %s add up to 100. The same goes for nurses and doctors. Yet the table presents them all as if they’re related.
ANSWER: Tables have been adapted according to Reviewer suggestions. The foot notes in all tables include the acronym explanation to help readers. In addition, horizontal lines have been added in Tables 1, 2 and 4 to separate unrelated variables.
- The data collection is confusing. The authors discuss DHEA as part of their own recent publications, but this has nothing to do with the study as it is presented. This is not part of the inclusion criteria, the method, or the analyses.
ANSWER: We agree with the Reviewer that this introduction in the “Data collection” section fits much better at the beginning of Results. See lanes 165-175.
- It’s unclear why Python was required for such basic analyses. What was Python used for? R is more than capable of running t-tests and descriptive analyses.
ANSWER: We agree with the Reviewer in the fact that R is a powerful and sufficient enough statistical software. Nevertheless, in our case we needed to apply several data cleaning and formatting operations that are easier to perform with Python (provided experience with the language). On the other hand, the statistical packages in Python can provide (in our opinion) more versatile graphs.
- What standard are the authors using to define low, medium, and large effect sizes? There are guidelines in the literature.
ANSWER: Reference values are indicated. See foot notes in Table 5 and new Reference 41.
- Some of the presented findings do not seem to correspond to any hypothesis. Why are there comparisons between professions, hospitals, and gender? The introduction does not lead the reader to expect these analyses.
ANSWER: It is true that we have not formulated a preliminary hypothesis. However, the presentation of data in this way gave as key finding for nurses/women. This observation motivate our team to perform a dynamometry test in this particular segment and the results will be published in a subsequent report, but we need to present first these data. We have observed that nurses that are women undergo more fatigue at the end of the working way which is reflected by significant changes in the strength test. These changes seem to correlate with the scores observed in the different psychological tests. The manuscript is almost finished but we need to publish first the present report in order to advance in our main project research.
Reviewer 2 Report
I recommend for publicationAuthor Response
Thank you very much
Reviewer 3 Report
The review is in the attachment.

Author Response
REVIEWER-3
Thank you for opportunity to review the article entitled: Health and Quality of Life of Healthcare Professionals Working in Hospital Emergency Departments.
The work has many shortcomings and requires a thorough improvement.
My suggestions for Authors:
Abstract
According to instructions to authors Abstract should be without headings.
ANSWER: Headings have been eliminated from the Abstract according to Reviewer suggestions.
Introduction
The paper was examined PQOL's professional quality of life (QPL-35), not QOL overall quality of life therefore, please consider changing the title of the work and address this issue both for the purpose and throughout the work.
ANSWER: Changes have been performed in the manuscript according to Reviewer suggestions. See lanes 32, 137, 140, 197, 202 and 212.
Aim of the paper
In the Abstract…...to complete this research, we decided to evaluate the quality of life of health professionals working in the Emergency Departments of two Spanish Hospitals.
In the Introduction…… aim of this work is to complete the aspects related with stress studied in previous reports [20,21] with the quality of life perception of healthcare workers…
The aim should be the same in all the work.
ANSWER: The aim is the same in the manuscript. This misinterpretation in the Abstract is due to word limitations imposed by the Instructions. The sentence in the Abstract should be interpreted as follows: To complete this research (aspects related to stress obtained from previous publications) we decided to evaluate the quality of life of health professionals working in the Emergency Departments of two Spanish Hospitals. A similar sentence is stated then in the Introduction where word limitation does not exist (see lane 99).
Line101 – (ED) but first was used in the abstract
ANSWER: The acronym is not used in the Abstract. The first time that it is used is in lane 101.
Material and methods
1.1 Study design.
Line 111 – (CEIC) please, explain the abbreviation (line 40).
ANSWER: CEIC is the Spanish acronym of the Ethics Committee (CEIC: “Comité Ético de Investigación Clínica”). We have explained de acronym in lane 117.
2.2 Sample
The basic criterion for inclusion was occupation and place of work – please complete.
ANSWER: The criterion has been included. See line 120.
Table 1. Reword: age, gender, profession, place of work.
Below the table explain the abbreviations HCUV and HSBS.
ANSWER: The suggested words have been included in the new version of Table 1. Abbreviation used are indicated in foot notes of the Table.
Table 2. Ages of participants - What do these results contribute to the study?
Maybe would be possible to merge Tables 1 and 2.
ANSWER: We have evidence that other research groups are doing complementing studies. They indicated that age could be an interesting variable to be taken into account. For this reason, we show the data regarding the age in a particular Table (Table 2). One the problems of the Spanish Health System is the generational renewal. Researches in this field need to know if age could be considered a conditioning variable to perform an optimal work in Emergency Departments. We hope that our data could contribute to increase the knowledge in this field.
2.3 Data collection
In many places, the authors refer to previously published results. I suggest that this information be included in the introduction at the end before the introduction. E.g.
The results presented in the paper are part of a larger project (title) and complement the previously published results [20,21].
The Authors refer to their two previous publications several times!
ANSWER: The suggested sentence has been added at the end of Introduction. We think that is good to refer to our previous publications to place the reader in the context of our research. See lanes 103-105.
Tool
Some information was missing:
Who is the author of the questionnaire? Please enter in the text and indicate a footnote.
ANSWER: The authors of all questionnaires are indicated in References 33-40.
Too laconic description of the tool – in the abstract there are 3 areas: 3 dimensions of the questionnaire: managerial support, workload and intrinsic motivation. There is no such information in this part of the work – please describe the tool in detail. And how the results were calculated (key).
What was reliability of the tool? (Eg. PMID: 18620633)
ANSWER: The questionnaire is largely explained in Section 2.3 (lanes 137-158). The questionnaire was reliable according to PMID: 18620633 (Reference 34).
Moreover
It is worth supplementing the information on the course of the study, e.g. Who handed out the questionnaires? How long did it take to fill?
ANSWER: The person that handed the questionnaire is indicated in the section “Author Contributions”. This person was the first author: Daniel Pérez-Valdecantos. This information is indicated as “investigation, D.P.-V.”. See lanes 304-305. The average time to fill the questionnaire is indicated in Section 2.3 (lane157-158).
Lines -120 – 38 medical professionals or medicine doctors?? Please specify.
ANSWER: The correct is “Medicine Doctors”. This has been corrected in lanes 33 and 120.
Lines - 135-139 please add Authors and footnotes at all scales.
ANSWER: According to “Instructions for Authors” we have to indicate the reference. See new References 33-40.
Lines 33, 137, 140 – please stick to English abbreviations in all work (e.g. QPL not CVP).
ANSWER: The English acronym has been used in the text (See lanes 32, 137, 140, 197, 202 and 212). The only exception is in Section 2.3 (lanes 138-139) because we have to explain to the readers that we passed the Spanish version of QPL-35, called CVP-35 which is validated as well (Reference 33).
Line 140 – is footnote [1] correct?
ANSWER: We cited the References 33 and 34.
Line 84 and throughout the discussion, please apply the correct footnote style, without indicating the year in parentheses.
ANSWER: According to Instructions for Authors, the best is to cite the corresponding reference. Therefore, we changed throughout the manuscript. See lanes 83, 243-244, 247-248 and 262-263.
Fig 1. “Medicine” should be Medicine doctors, “nursing” – Nurses, “Category” – profession.
ANSWER: Figure 1 has been changed accordingly.
Line - 184 - there is abbreviation QOL, but this term appears in the text for the first time in line 48, please include this abbreviation there.
ANSWER: This abbreviation refers to the score of QPL-35. This has been changed in Figure 1 and in the legend of the figure.
Discussion
Line 259 - It is true that some research carried out in similar populations, Which studies? No footnotes.
ANSWER: We have indicated some References (20-30).
Limitations - In my opinion small sample was one of it.
ANSWER: This limitation has been included (lane 289).
Conclusions
Refer to the results - professional quality of life PQOL.
ANSWER: Conclusions have been changed accordingly (see lanes 296-301).
Round 2
Reviewer 1 Report
The authors have addressed my concerns and have made the required changes. I appreciate their thoughtful replies. Great work!
Reviewer 3 Report
In light of the explanations and changes made I recommend the paper for publication.